# PPARγ in Atherosclerotic Endothelial Dysfunction: Regulatory Compounds and PTMs

**DOI:** 10.3390/ijms241914494

**Published:** 2023-09-24

**Authors:** Jinwen Luan, Xiaohui Ji, Longhua Liu

**Affiliations:** School of Exercise and Health, Shanghai University of Sport, Shanghai 200082, China

**Keywords:** endothelial dysfunction, PPARγ, atherosclerosis, post-translational modifications, TZDs

## Abstract

The formation of atherosclerotic plaques is one of the main sources of cardiovascular disease. In addition to known risk factors such as dyslipidemia, diabetes, obesity, and hypertension, endothelial dysfunction has been shown to play a key role in the formation and progression of atherosclerosis. Peroxisome proliferator-activated receptor-gamma (PPARγ), a transcription factor belonging to the steroid superfamily, is expressed in the aorta and plays a critical role in protecting endothelial function. It thereby serves as a target for treating both diabetes and atherosclerosis. Although many studies have examined endothelial cell disorders in atherosclerosis, the role of PPARγ in endothelial dysfunction is still not well understood. In this review, we summarize the possible mechanisms of action behind PPARγ regulatory compounds and post-translational modifications (PTMs) of PPARγ in the control of endothelial function. We also explore the potential use of endothelial PPARγ-targeted agents in the prevention and treatment of atherosclerosis.

## 1. Introduction

Endothelial dysfunction plays important roles in atherosclerosis initiation and progression. Atherosclerosis poses a major threat to human health and is one of the leading causes of morbidity and mortality in type 2 diabetes (T2DM) patients [1,2]. Much evidence suggests that atherosclerosis is initiated by flow disturbance-stimulated endothelial activation, followed by subendothelial apolipoprotein B (apoB) retention, monocyte-derived macrophage entry, smooth muscle cell (SMC) proliferation, and plaque formation [3,4]. Atherosclerosis is a lipid-driven, multifocal, chronic immunoinflammatory disease [5,6], with clinical outcomes (e.g., heart attack and stroke) that are primarily attributable to plaque rupture with thrombosis. In addition, vascular occlusion caused by narrowing of blood vessels and platelet aggregation can lead to acute cardiovascular events [7]. Endothelial activation is critical in the initiation of atheroma formation [8,9]. Endothelial cells (ECs) exist in various tissues and organs in humans and are indispensable in the regulation of vasodilation, SMC proliferation, and the inflammatory response [10]. In fact, the intact endothelium promotes anticoagulation and fibrinolysis [8] and contributes to vascular homeostasis [11]. Endothelial-dependent vasodilation disorders (caused by defects in nitric oxide (NO) production or activity [10]) and maladaptation of adhesion molecule expression [12] play critical roles in the development of atherosclerosis in both humans and mice [1,10]. Fortunately, EC dysfunction appears to be reversible [8].

Peroxisome proliferator-activated receptor gamma (PPARγ), a nuclear receptor of the steroid superfamily, is a potent target for counteracting risk factors of atherosclerosis. Thiazolidinediones (TZDs), synthetic ligands of PPARγ, have been shown to reduce the incidence of atherosclerosis [13]. PPARγ has two major isoforms: PPARγ 1 and PPARγ 2. PPARγ 1 is expressed in a variety of tissues including SMCs and ECs that participates in the atherogenic process, while PPARγ 2 is expressed only in adipose tissue, contains an additional 30 amino acids in its N-terminal region [14,15]. As a transcription factor, PPARγ is involved in regulating glucose and lipid metabolism [15,16] (by promoting fat formation and increasing insulin sensitivity), endothelial function [17], and inflammation [18,19,20]. Additionally, recent studies have shown that PPARγ has pleiotropic effects and is important in maintaining cardiovascular and renal function [21,22].

PPARγ can be regulated by altering the activity of PPARγ or modulating its downstream pathways. With regard to the former, the bioactivity of PPARγ can be regulated by ligands that bind to peroxisome proliferator response elements (PPREs), the corepressor complex, and the co-activator complex through different domains [14,23]. PPARγ contains four different domains: ligand-independent activation function 1 (AF1), DNA-binding domain (DBD), ligand-binding domain (LBD), and activation function 2 (AF2) [14,24]. PPARγ can bind to PPREs of target genes through its DBD [14]. Moreover, via the augmentation of co-repressor complexes, a process termed trans-repression, unliganded PPARγ may exert anti-inflammatory effects [23]. In contrast, ligand-bound PPARγ forms the coactivator complex [14]. With regard to downstream pathway modulation of PPARγ, its post-translational modifications (PTMs) extensively affect its protein stability, trans-activation function, and cofactor interaction [25]. PTM patterns include acetylation, phosphorylation, ubiquitination, and small ubiquitin-related modifier (SUMOylation). In addition to the mRNA expression of *PPARγ*, increasing attention has been directed towards these PTMs [24,25,26,27]. In this review, we summarize the normal function versus dysfunction of ECs, with an emphasis on regulatory compounds and PTMs of PPARγ. Analysis of PPARγ-interacting compounds may identify potential drug targets for novel therapeutics, and investigation of PTMs may inspire new research directions in the future.

## 2. Endothelial Function and Atherosclerosis

### 2.1. Endothelial Function and Associated Molecular Pathways

The cardiovascular system is responsible for distributing nutrients and oxygen throughout the body [28], and one of the most important contributors to vascular homeostasis is the endothelium [29]. ECs comprise the innermost layer of the luminal surface of blood vessels, facilitating fluid flow and material exchange [30]. Although no longer proliferating in adults, ECs constantly receive and activate signals that direct their activity [31]. The endothelium is more than just a simple membrane or barrier [30,32]; as a complex autocrine, paracrine, and exocrine organ, the endothelium regulates luminal pressure by releasing factors and responding to external stimuli [31]. The major functions of the endothelium are summarized in Figure 1.

In terms of physiologic function, the endothelial barrier is responsible for the exchange of substances entering or exiting the blood [33,34]. The glycocalyx is a polysaccharide-protein complex expressed on the endothelial cell surface that reduces blood flow resistance and blood vessel adhesion. Damage to the glycocalyx and inflammation alter the configuration of ECs and change endothelial permeability via intracellular factors [34,35]. Moreover, ECs contribute to both the innate and adaptive immune responses [36] (Figure 1).

ECs also play critical roles in maintaining the appropriate hemostasis by interacting with coagulation and fibrinolytic system [37]. Activation of proteins involved in the anticoagulation pathway limit thrombosis [38], while secretion of endothelial adhesion molecules and thrombomodulin repair vascular injury [39]. Loss of thrombomodulin secretion by ECs has been associated with the side effects of allotransplantation, mediated by the inflammatory factor tumor necrosis factor-α (TNF-α) [40,41]. T2DM and inflammation can also stimulate endothelial coagulation and activate platelets [42]. When the activity of NO is reduced and the activity of NO synthase (NOS) is inhibited, acute deterioration of endothelial function occurs [43]. Supplementation with L-arginine can improve endothelium-dependent dilatation [44]. Additionally, vascular endothelial growth factor (VEGF) and angiogenin (Ang)-Ⅱ expression can increase endothelial permeability, while Ang-Ⅰ has the opposite effect [45]. High permeability can be induced by mechanical stretch via deacetylation of endothelial α-tubulin [46].

Similarly, certain substances in the blood are involved in regulating the vascular function of endothelium. For example, high levels of Thyroid Stimulating Hormone (TSH) in blood can trigger excessive opening of the mitochondrial permeability transition pore as well as increased mitochondrial oxidative damage in ECs [47]. Blocking this activation can ameliorate endothelium-dependent vasodilation disorders [47]. Acute vasodilation can also be induced by high levels of free ferulic acid (FA), found in high-fiber bread [48]. Excess circulating lipids and cholesterol (especially LDL cholesterol) infiltrate ECs. They subsequently cause endodermal inflammation, alter endothelial permeability, drive chemokine secretion, and recruit monocytes, all of which are key mechanisms of early atherosclerosis formation.

ECs secrete connective tissue components, collagen molecules, proteoglycans, and elastin to regulate cell-to-cell or matrix binding, which plays an important role in the stability of early plaques. However, with progressive age, rupture of collagen fibers and degradation of the extracellular matrix occurs, resulting in plaque rupture and subsequent cardiovascular events. Reducing monocyte adhesion to the aortic endothelium and modulating levels of vascular cell adhesion molecules have demonstrated beneficial effects [49,50].

The function of ECs can be evaluated by testing the impairment of vascular dilation with acetylcholine [51,52], and new molecular imaging techniques have made it easier to assess lesions in real time [53]. Analysis of inflammatory factors, oxidation markers, reactive oxygen species (ROS), cell proliferation, and apoptosis also aid in evaluating EC function [52,54].

### 2.2. Endothelial Dysfunction and Its Role in Atherosclerosis Development

Atherosclerosis is widely recognized as a chronic inflammatory disease [55,56]. Plaques containing a large necrotic core and thin fibrous cap, along with subsequent rupture and erosion of the plaque surface, cause acute cardiovascular or cerebrovascular events [57,58]. Plaque rupture is attributed to structural instability of the plaque, while surface erosion appears to be caused by endothelial apoptosis and detachment [58,59]. Effective treatment methods include lipid modulators and anti-inflammatory medications. Specifically, statins improve lipid profiles, although residual cardiovascular disease risk remains [60,61]. Anti-inflammatory medications reduce this “residual risk” independently of lipid modification, and thereby further reduce the incidence of cardiovascular events [62].

The term “endothelial dysfunction” can be used to refer to the impairment of vascular function caused by abnormal production or activity of endothelium-derived NO [10]. Here, however, we use the term to refer to endothelial dysfunction in atherosclerosis more broadly. Per the review by Gimbrone Jr. and García-Cardeña in 2016 [8], endothelial dysfunction encapsulates changes in endothelial adaptation associated with atherosclerotic cardiovascular disease. Such dysfunction broadly manifests as intravascular imbalances in hemostasis and thrombosis, increased local vascular tone, redox imbalance [63], and both acute and chronic inflammatory responses [8] (Figure 1). For example, high thrombomodulin concentration and circulating EC level suggest the presence of an endothelial lesion [64], while oxidative stress is associated with hyperglycemia-induced endothelial dysfunction [65]. Vascular endothelial dysfunction is a marker of atherosclerosis [9] and peripheral circulatory diseases [66]. In fact, endothelial dysfunction in susceptible locations within arteries is the earliest detectable change in atherosclerosis, while advanced plaques have surrounding inflammatory cells that induce the pro-inflammatory phenotype [8]. Many clinically used anti-atherosclerotic compounds function in part by counteracting EC dysfunction [9].

## 3. Effects of PPARγ on Endothelial Function

### 3.1. PTMs of PPARγ

PPARγ has several PTM patterns, including phosphorylation, SUMOylation, and acetylation [25]. These PTMs can significantly alter its transcriptional activity [67], thereby inducing selective activation. The modification sites of each isoform of PPARγ are slightly different. In mice, the modification sites for PPARγ 2 are Y78, K98, K107, S112, S133, K218, K268, S273, K293, and S296, while the modification sites for PPARγ 1 are K63, T84, K94, K98, K184, K185, and K395 (K—lysine, S—serine, Y—tyrosine, T—threonine) [25]. In ECs, the PTM sites for PPARγ are Lys77/107, Lys268, Lys293, and Lys462 (Figure 2).

### 3.2. Molecular Pathways of Endothelial Protection by PPARγ

PPARγ is involved in several molecular pathways that promote endothelial function (Figure 3), as outlined in the following paragraphs.

Phosphorylation and inactivation of PPARγ is directly catalyzed by extracellular regulated protein kinase 1/2 (ERK1/2), a pro-inflammatory kinase [20]. Furthermore, phosphorylation of ERK1/2 has an activating effect on PPARγ activity [68]. Upregulation of the CTRP6/ERK/PPARγ (CTRP6, C1q/tumor necrosis factor-related protein 6) axis has been shown to promote primary aortic endothelial cell proliferation and migration [68].

NO production and eNOS activity also play integral roles in the effect of PPARγ on endothelial function [52,69,70,71]. The Akt/eNOS/NO cascade regulates endothelial-dependent relaxation of blood vessels [72] (Figure 3). PPARγ may facilitate the release of NO through its downstream PI3K/Akt (PI3K, phosphatidylinositol 3-kinase) pathway [69,72]. In human glomerular ECs, activation of nuclear factor erythroid 2-related factor 2 (Nrf2), which promotes the PPARγ/eNOS pathway, is involved in NO production [73].

PPARγ participates in the recruitment of leukocytes; in particular, recruitment of monocytes to activated vascular ECs is a critical step in the inflammatory cascade [74]. PPARγ activation controls mononuclear attachment [75,76] and thus has anti-inflammatory properties via inhibition of protein N-glycosylation in adhesion molecules [77]. Activation of the PPARγ/LXRα/ABCA1 pathway may downregulate ICAM-1, VCAM-1, interleukin (IL)-6, and TNF-α [75]. Blood flow also affects the recruitment response, and activation of PPARγ is a key mediator of flow-dependent monocyte adhesion [74].

NF-κB is involved in PPARγ-mediated endothelial protection [78,79,80,81,82,83]. IκBα degradation and NF-κB p65 nuclear translocation may be mediated by lipopolysaccharides [79] and inflammatory factors. These two processes are weakened by PPARγ activation. Western blotting demonstrated reduced phosphorylation of NF-κB p65 with PPARγ activation [79]. The PPARγ-NADPH oxidase/ROS-NF-κB pathway may be involved in protecting the endothelium from H_2_O_2_-induced injury [84]. Susceptibility to insulin resistance (IR) can be reduced by inhibiting the PPARγ-dependent NF-κB pathway [78].

Lectin-like receptor for oxidized LDL (LOX-1), located on the membrane of ECs, can recognize oxLDL, TNFα, and other stimuli, and plays an important role in the progression of atherosclerosis [85,86]. In bovine aortic ECs, PPARγ ligands decreased LOX-1 expression induced by TNFα [87]. PPARγ activation also inhibited LOX-1 and the adhesion of monocytes to oxLDL-treated human coronary artery ECs by suppressing superoxide radical generation [88].

Other mechanisms of endothelial protection by PPARγ have been identified, including its role in apoptosis. Upregulation of PPARγ was found to induce the upregulation of p53 and promote apoptosis of HUVECs [89]. Activation of PPARγ prompts the opening of maxi-K channels and subsequently inhibits angiogenesis by promoting apoptosis [90]. Nuclear PPAR-γ, but not cytoplasmic PPAR-γ, has been shown to protect against endothelial dysfunction via transcriptional activation of miR-590-5p in ECs [91].

Furthermore, gene knockout and overexpression techniques have facilitated research efforts into the endothelial-protective role of PPARγ. In a comparison between endothelial-specific and macrophage-specific PPARγ disruption in LDL receptor-knockout (Ldlr^−/−^) mice, endothelial PPARγ demonstrated anti-inflammatory and hypopermeability properties, thereby contributing to a protective effect against atherogenesis [92]. The endothelial-specific PPARγ dominant negative mutations markedly impaired vasodilatory responses to acetylcholine, which was subsequently restored by a scavenger of superoxide [93]. Moreover, overexpression of PPARγ may prevent endothelial injury in the aorta by reducing oxidative stress [94].

### 3.3. Compounds Regulating PPARγ and Their Role in Endothelial Function

PPARγ is regulated by endogenous or synthesized compounds that affect its activity. These compounds can be divided into the following categories: TZDs [71,81,82,90,95,96,97,98,99,100,101], alkaloids [102,103,104], saponins and triterpenes [83,105,106], isoflavones [54,74], biological metabolites [72,107], dual PPARα/γ agonists [108,109], and other compounds (Table 1). In this section, we discuss the potential mechanisms of these compounds in regulating endothelial function through PPARγ.

#### 3.3.1. TZDs

TZDs are full agonists of PPARγ and are used clinically as insulin sensitizers in the treatment of T2DM [110]. The most common TZDs include rosiglitazone [90,95,96,97], troglitazone [81,98], pioglitazone [99,100,101], ciglitazone [71,82], and lobeglitazone [111]. TZDs act through classical PPARγ-dependent insulin signaling pathways and improve insulin sensitivity by inhibiting nonclassical PPARγ-dependent NF-κB pathways [78]. In addition, TZDs may reduce EC activity by upregulating diacylglycerol kinase and downregulating the diacylglycerol-protein kinase C signaling pathway [76,96].

Many studies have focused on the TZD rosiglitazone. Rosiglitazone is a PPARγ ligand that promotes vascular protection through antioxidant and anti-nitrification effects [95]. The use of rosiglitazone may induce eNOS activity and NO production (Figure 3), thereby triggering apoptosis and inhibiting angiogenesis by increasing PPARγ-mediated maxi-K channel opening [90]. The pharmacological effects of rosiglitazone may also be associated with increased endothelial-dependent vasodilation [95], inhibition of leukocyte accumulation [95], and reduced inflammation and ERK activation [96,97]. Specifically, administration of rosiglitazone demonstrated anti-inflammatory effects in human ECs, with downregulated ERK activity in response to TNF-α and IFNγ [96]. Inhibition of the AT1-ROS-MAPK signaling pathway may also be involved in the anti-inflammatory effects of rosiglitazone (AT1: Ang II type 1 receptor, MAPK: mitogen-activated protein kinase) [97].

In contrast, there are relatively few studies involving other TZDs. Troglitazone may inhibit the ROS/NF-κB signaling pathway [81]; however, it has been reported that troglitazone-induced endothelial autophagy may be independent of PPARγ activation [98]. Pioglitazone has been shown to promote apoptosis and endothelial-dependent relaxation while inhibiting endothelial regeneration [99]. Its function may rely in part on VEGF/FGF (fibroblast growth factor) stimulation of the ERK 1/2 pathway [99]. Ciglitazone may protect against inflammation via the PPARγ/NF-κB pathway [82] and promote vasodilation through increased NO production independent of eNOS expression [71]. Finally, in a study conducted by Lim et al., lobeglitazone treatment was found to significantly reduce neointimal formation in rats, decrease the expression of monocyte adhesion molecules in vitro, and reduce NF-κB p65 nuclear translocation [111] (Table 1).

#### 3.3.2. Alkaloids

Hypaphorine and berberine are classified as alkaloids. Hypaphorine was found to reverse lipopolysaccharide-induced downregulation of endothelial PPARγ by regulating the PI3K/Akt/mTOR (mTOR: mammalian target of rapamycin) signal pathway, thus producing anti-inflammatory effects [103]. Berberine acts as a plaque stabilizer by promoting PPARγ expression and reducing oxidative stress in the endothelium in conditions of elevated plasma homocysteine levels [104].

#### 3.3.3. Saponins and Triterpenes

Notoginsenoside Fc, ginsenoside-Rb1, and 7,8-didehydrocimigenol (7,8-DHC) are members of the saponin and triterpene class. Notoginsenoside Fc was found to attenuate high glucose (HG)-induced EC injury through the inhibition of pro-inflammatory cytokines and apoptosis as well as increased PPARγ proliferation in rat aortic ECs [105]. Activation of the PPARγ signaling pathway was also involved in ginsenoside-Rb1-modulated anti-angiogenesis, through downregulation of miR-33a expression and upregulation of pigment epithelial-derived factor (PEDF) protein in HUVECs [106]. Lastly, 7,8-DHC extracted from a traditional herb selectively inhibited VCAM-1 but not ICAM-1 expression by increasing PPARγ levels and inhibiting NF-κB activity in TNF-α-activated human ECs [83].

#### 3.3.4. Isoflavones

Genistein and formononetin are examples of isoflavones. Genistein inhibits monocyte adhesion to the endothelium via PPARγ and the physical forces associated with blood flow, rather than by regulating the expression of adhesion molecules [74]. Activation of PPARγ signaling by formononetin reduces inflammation, oxidative stress, and apoptosis, thereby limiting endothelial damage in oxLDL-induced atherosclerosis [54].

#### 3.3.5. Biological Metabolites

2-methoxyestradiol and urolithin A(UA) are biological metabolites. 2-methoxyestradiol is an endogenous estradiol metabolite with antiatherogenic properties. This molecule has been found to promote the release of NO primarily through the PPARγ/PI3K/Akt/eNOS cascade, thereby encouraging endothelial-dependent vasodilation [72]. UA, an intestinal metabolite, has been shown to significantly suppress phosphorylated ERK1/2 and improve endothelial function partly through downregulation of microRNA-27 expression and upregulation of the ERK/PPARγ pathway [107].

#### 3.3.6. Dual PPARα/γ Agonists

Dual PPARα/γ agonists include TAK-559 and conjugated linoleic acid [108,109]. TAK-559 inhibits the recruitment of macrophages and vascular SMCs, which in turn may lead to a decrease in intimal thickness [108]. Conjugated linoleic acid has demonstrated anti-atherosclerotic effects in coronary artery ECs by normalizing gene expression profiles induced by hemodynamic features of the circulation [109].

#### 3.3.7. Others

This category consists of compounds that do not fall into any of the aforementioned groups.

15-deoxy-Delta (12,14)-prostaglandin J2 (15d-PGJ2) is a prostaglandin that is a strong, naturally existing PPARγ agonist [112]. 15d-PGJ2 stimulated NO release in vascular ECs through an eNOS expression-independent transcriptional mechanism [71]. 15d-PGJ2 may also act through the diacylglycerol-protein kinase C signaling pathway [76] and the NF-κB pathway [81,82].

Telmisartan, an angiotensin II-receptor blocker (ARB), was recently reported to have PPARγ partial agonist properties [52,70]. Telmisartan protects EC function [52,69] and improves cardiovascular remodeling [70]. In a monocrotaline-induced pulmonary hypertension model, treatment with telmisartan protected endothelial cell function by inducing the expression of PPARγ, promoting the phosphorylation of Akt and eNOS, increasing the production of NO, and improving PPARγ-dependent vascular dilation [69]. In the Dahl salt-sensitive hypertensive rat model, telmisartan treatment exhibited cardioprotective effects including reduced oxidative stress and cardiovascular lesion formation [70].

KR-62980, a new PPARγ agonist, impaired angiogenesis in HUVECs and promoted apoptosis through PPARγ-mediated downregulation of VEGF receptor 2 (VEGFR-2) and upregulation of chromosome 10 (PTEN) [113].

Tert-Butylhydroquinone is an activator of Nrf2 that may increase the expression of eNOS by up-regulating the transcription and translation of PPARγ, subsequently increasing endothelial NO production [73].

Magnolol is a dual agonist of RXRα and PPARγ. In cultured HUVECs, incubation with magnolol enhanced the release of NO and promoted insulin-induced vasodilation through the PPARγ-dependent Akt/eNOS/NO cascade [114].

Cyclic phosphatidic acid (cPA) potentially inhibits neointima formation by attenuating VEGF-mediated growth and migration through upregulation of the cPA-PPARγ axis in human coronary artery ECs from patients with diabetes [115].

1,8-Cineole may protect the endothelium from inflammatory infiltration by upregulating PPARγ expression and inhibiting NF-κB p65 phosphorylation [79].

Carbon monoxide derived from CO-releasing molecule-2 (CORM-2) exhibited anti-atherogenic effects through attenuation of HG-induced endothelial ICAM-1 via the AMPK/PPARγ pathway [116].

Simvastatin, a member of the statin class, demonstrated benefits in maintaining thrombotic homeostasis of saphenous vein ECs challenged by advanced glycation endproducts. Decreased PPARγ expression was significantly improved by simvastatin in saphenous vein ECs from non-diabetic patients in vitro. There was also a slight, but not statistically significant, increase in PPARγ expression in ECs obtained from diabetic patients treated with simvastatin [117].

**Table 1 ijms-24-14494-t001:** Compounds regulating PPARγ and their role in endothelial function.

Class	Compound	Endothelial Change	Mechanism	References
TZDs	rosiglitazone	↓oxidative and nitrative stresses, angiogenesis, leukocyte accumulation, inflammation ↑endothelium-dependent vasodilation, apoptosis	↑PPARγ expression, eNOS activity, NO production ↓gp91phox and iNOS expression, superoxide and total NOx, nitrotyrosine, ERK activation ↑PPARγ-mediated maxi-K channel opening ↓diacylglycerol-protein kinase C signaling pathway ↓AT_1_-ROS-MAPK signal pathway	[90,95,96,97]
	troglitazone	↑Autophagy	independent of EGFR transactivation and PPARγ activation ↓ROS/NF-κB signaling pathway	[81,98]
	pioglitazone	↓endothelial regrowth ↑apoptosis, endothelium-dependent dilation	↑eNOS activity, VEGF protein levels, p38 MAPK activation ↓ERK activation ↓VEGF/FGF stimulation of the ERK 1/2 pathways	[99,100,101]
	ciglitazone	↓inflammation ↑vasodilation	↓NF-κB pathway through a PPARγ-dependent mechanism ↑NO independent of eNOS expression	[71,82]
	lobeglitazone	↓leukocyte recruitment, inflammation, intima-media ratio	↓adhesion molecules, NF-κB p65 nuclear translocation	[111]
Alkaloids	hypaphorine	↓inflammation	↓TNF-α, IL-1β, MCP-1, VCAM-1 and TLR4 ↑PPARγ protein levels, phosphorylation of AMPK and ACC, Akt and mTOR ↓TLR4 and ↑PPARγ, dependent on PI3K/Akt/mTOR signal pathway	[102,103]
	berberine	↓oxidative stress ↑vasodilation, cell viabilities	↑PPARγ ↓ROS	[104]
Saponins and triterpenoids	notoginsenoside Fc	↓inflammation, apoptosis ↑proliferation	↑PPARγ ↓pro-inflammatory cytokines	[105]
	ginsenoside-Rb1	↓angiogenesis	↑PPARγ, PEDF protein ↓miR-33a expression	[106]
	7,8-didehydrocimigenol (7,8-DHC)	↓leukocyte recruitment, inflammation	↑PPARγ ↓VCAM-1 (but not ICAM-1), NF-kB activity	[83]
Isoflavones	genistein	↓leukocyte recruitment, inflammation	↑PPARγ Dependent on flow rather than regulation of the adhesion molecules	[74]
	formononetin	↓inflammation, oxidative stress, apoptosis	Stimulates PPARγ signaling	[54]
Biological metabolites	2-methoxyestradiol	↑vasodilation	↑p-Akt, p-eNOS, NO Via the PPARγ/PI3K/Akt pathway	[72]
	Urolithin A(UA)	↓monocyte adhesion, inflammation	↓microRNA-27 expression ↑ERK/PPARγ pathway	[107]
Dual PPARα/γ agonists	TAK-559	↓leukocyte recruitment, intimal thickness	↓*MCP-1* mRNA expression and secretion	[108]
	conjugated linoleic acid	Normalizes EC responses to hemodynamic change	↑PPAR and eNOS expression ↓Pro-atherogenic ET-1 response	[109]
Others	15d-PGJ2	↓endothelial cell activation, inflammation ↑vasodilation	↑NO independent of eNOS expression ↑diacylglycerol kinase ↓diacylglycerol-protein kinase C signaling pathway ↓proinflammatory adhesion molecule expression ↓NF-κB pathway through a PPARγ-independent mechanism ↓ROS/NF-κB signaling pathway	[71,76,81,118]
	tert-Butylhydroquinone	↑vasodilation	↑HO-1 to ↓ROS ↑DDAH to ↓asymmetric dimethylarginine ↑PPARγ/eNOS	[73]
	Magnolol	↑insulin-induced vasodilation	↑PPARγ, insulin-induced phosphorylated Akt and eNOS levels, NO production ↓TRB3	[114]
	Telmisartan	↑vasodilation ↓medial thickness, oxidative stress	↑PPARγ, eNOS, p-eNOS, p-Akt, NO production ↓nitrotyrosine Via a PPARγ-dependent PI3K/Akt/eNOS pathway Related to the PPARγ/eNOS pathway and Rho-kinase pathway	[52,69,70]
	Cyclic phosphatidic acid (cPA)	↓neointima formation	↓VEGF-mediated growth and migration ↓PPARγ, cPA-PPARγ axis	[115]
	1,8-Cineole	↓inflammation	Via the PPARγ/NF-κB pathway ↑PPARγ expression ↓VCAM-1, phosphorylation of NF-κB p65, E-selectin, IL-6, and IL-8	[79]
	carbon monoxide	↓leukocyte recruitment	↓endothelial ICAM-1 ↑AMPK/PPARγ pathway	[116]
	simvastatin	↓leukocyte recruitment	↓Neutrophil adhesion, ROS associated with ↑PPAR-γ and ↓RAGEs expression	[117]
	KR-62980	↓angiogenesis, EC proliferation and chemotactic migration ↑apoptosis	↓VEGFR-2 ↑chromosome 10 (PTEN)	[113]

↑, increase; ↓, decrease.

### 3.4. PTMs of PPARγ Responsible for Endothelial Dysfunction

PTMs of PPARγ play critical roles in finely regulating its activity. PTMs of PPARγ, such as acetylation, phosphorylation, and SUMOylation, are known to regulate glucose and lipid metabolism [24,27]. Here, we will focus on the role of PTMs of PPARγ in endothelial dysfunction.

#### 3.4.1. Acetylation

PPARγ acetylation regulates PPARγ activity, and the dynamics of acetylation are disturbed in obesity and aging [119]. PPARγ is deacetylated by NAD-dependent deacetylase sirtuin 1 (SirT1), which may be activated by TZDs such as rosiglitazone [120]. SirT1 regulates endothelial function via several targets, including eNOS, FoxOs, and PPARγ [121]. Acetylates, such as CBP/P300 (CBP, CREB-binding protein), directly bind to PPARγ in at least two different sites and play an indispensable role in adipocyte differentiation [122,123]. Deacetylation of two lysine residues (K268, K293) on PPARγ was found to promote insulin sensitivity and browning of white adipose tissue, in addition to alleviating western diet-induced atherosclerosis in Ldlr^−/−^ mice [120,124]. This protective effect against atherosclerosis can partly be attributed to inhibition of the expression of iNOS (inducible nitric oxide synthase), Nox2 (NADPH nitric 2), and IL-6 in aortic endothelium in vivo and in primary ECs in vitro [125]. PPARγ deacetylation at K268 and K293 has also been shown to significantly increase acetylcholine-induced relaxation from phenylephrine-induced contraction [125]. Administration of deacetylase inhibitors may reverse the increased permeability of vascular trans-endothelial cells and the downregulation of tight junction protein expression [126]. As such, acetylation is a major contributor in accelerating endothelial inflammation in the development of atherosclerosis.

#### 3.4.2. SUMOylation

SUMOylation is another PTM that can selectively regulate PPARγ activity. PPARγ-K107R-mutant mice (K107R prevents SUMOylation) were found to have increased insulin sensitivity without the associated increase in adiposity that typically accompanies TZD treatment [127]. DeSUMOylation of PPARγ may inhibit neointimal formation and reduce vascular SMC proliferation, suggesting that it confers protection from atherosclerosis [128]. However, there is little research about the role of K107 deSUMOylation in regulating EC function specifically. SUMOylation at K77, which interacts with FoxO1, might mediate endothelial IR induced by high glucose and palmitic acid in HUVECs [129,130]. Additionally, mutations in the SUMOylation site of PPARγ2 in the AF1 region inhibits iNOS activity, resulting in beneficial effects on macrophage inflammation-associated genes [131]. Further analyses of its regulatory pathway suggest that SUMOylation of PPARγ may, to a large extent, promote the secretion of chemokines by ECs through oxidative stress and inflammatory pathways, thereby contributing to atherogenesis. Further studies using transgenic mouse models with PPARγ SUMOylation or deSUMOylation may provide additional evidence of this PTM’s role in regulating EC function and atherosclerosis development.

## 4. Conclusions and Future Directions

Given the importance of PPARγ in EC dysfunction and the role of EC disorders in atherosclerosis progression, it is essential that we understand the mechanisms behind PPARγ signaling in endothelial function. This can serve as a springboard for the development of novel PPARγ-targeted drug therapies in the future. Furthermore, understanding how PPARγ-modulating compounds and PTMs alter EC function may aid in identifying new therapeutic targets, such as deacetylation or deSUMOylation processes.

A PTM we did not cover in this review is PPARγ phosphorylation, as few studies have been conducted on this topic and its significance remains unclear. Based on existing studies, PPARγ phosphorylation at various sites has demonstrated a range of glucose and lipid metabolic effects. For example, phosphorylation at S273 by cyclin-dependent kinase 5 (CDK5) has been shown to increase proinflammatory factors including TNF-α, IL-1β, and foam cell formation, which worsen atherosclerosis [132]. Both PPARγ partial agonist SR1664 and antagonist SR10171 may inhibit phosphorylation at S273, resulting in increased insulin sensitivity [133,134]. Additionally, MAPK can phosphorylate PPARγ at both S273 and S112, and its pathway is activated by growth factors such as EGF and platelet-derived growth factors (PDGF). This was found to promote foam cell formation when exposed to oxLDL [135]. Nonetheless, further studies are needed to gain a more definitive understanding of the phosphorylation of PPARγ in ECs. Whether this PTM regulates EC function is worthy of additional investigation.

To conclude, we can reasonably anticipate: (1) The use of next-generation sequencing technologies, such as RNA sequencing, and other omics techniques to further evaluate the molecular basis of PPARγ regulation of EC dysfunction in the context of atherosclerosis. This will allow us to establish a more comprehensive regulatory network incorporating upstream and downstream targets of PPARγ. (2) There is growing evidence that selective activation of PPARγ may reveal novel therapeutic targets for future generations of drugs in the treatment of both T2DM and atherosclerosis, and PTMs of PPARγ have shown great potential in this regard. (3) Given the many different PTMs and regulatory sites of PPARγ, along with their apparent involvement in EC dysfunction, increasing attention will continue to be directed towards PPARγ regulation in other metabolic and cardiovascular diseases.

## Figures and Tables

**Figure 1 ijms-24-14494-f001:**
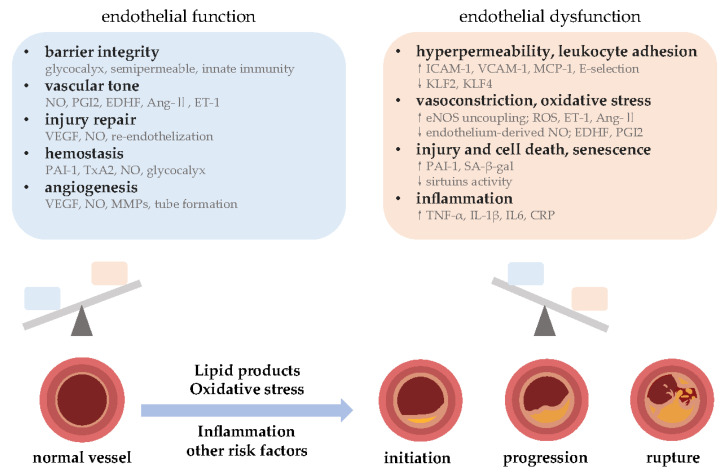
Normal endothelial function vs. dysfunction, and atherosclerosis development. Endothelial dysfunction affects atherosclerosis in many ways, as it is involved in endothelial integrity, vascular tone, injury repair, hemostasis, and angiogenesis. The functional state of the endothelium is dynamically regulated, and molecular expression in the endothelium changes with high levels of lipid production, oxidative stress, inflammation, etc., resulting in increased permeability, leukocyte adhesion, vasoconstriction, and apoptosis. These collectively contribute to the development of atherosclerosis. Abbreviations: PGI2, prostacyclin; EDHF, endothelium-derived hyperpolarizing factor; Ang-II, angiotensin II; ET-1, endothelin 1; VEGF, vascular endothelial growth factor; PAI-1, plasminogen activator inhibitor-1; TxA2, thromboxane A2; MMPs, matrix metalloproteinases; ICAM-1, intercellular adhesion molecule-1; VCAM-1, vascular adhesion molecule-1; MCP-1, monocyte chemoattractant protein-1; KLF, Kruppel-like factor; eNOS, endothelial nitric oxide synthase; SA-β-gal, senescence-associated β-galactosidase; TNF-α, tumor necrosis factor α; IL, interleukin; CRP, c-reactive protein; NO, nitric oxide; ROS, reactive oxygen species.

**Figure 2 ijms-24-14494-f002:**
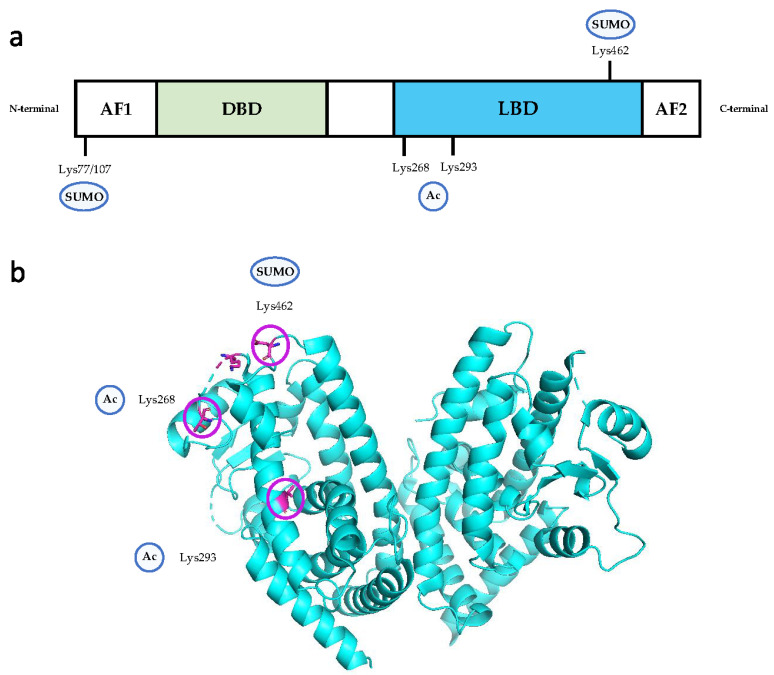
The post-translational modification sites of PPARγ in endothelial cells. The acetylation sites are Lys268 and Lys293 in the LBD region, and the SUMOylation sites are Lys462 in the LBD region and Lys77/107 in the AF1 region. (**a**) Structural characteristics of PPARγ with PTM sites identified. (**b**) Three-dimensional representation of the distribution of PTM sites described in (**a**), using a ligand-free model of the human PPARγ LBD. (PDB DOI: 10.2210/pdb6L8B/pdb. NDB: 6L8B). The PTM sites are emphasized in the form of purple circles. Abbreviations: Ac, acetylation; SUMO, SUMOylation.

**Figure 3 ijms-24-14494-f003:**
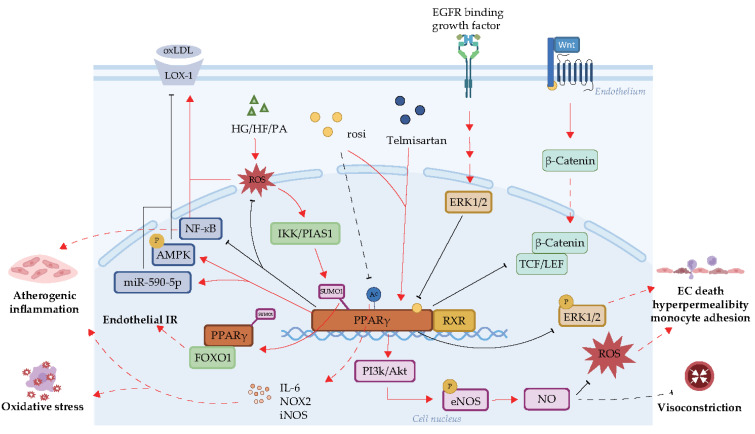
Interacting compounds and PTMs of PPARγ that regulate its function in the endothelium. Activation of PPARγ itself inhibits phosphorylation of ERK1/2 and promotes the PI3K/AKT/eNOS/NO cascade. These two pathways collectively inhibit EC death, decrease leukocyte adhesion to the activated endothelium, and reduce permeability of the arterial wall. The upregulation of PPARγ can also inhibit the expression of NF-kB and reduce atherosclerotic inflammation. PPARγ has a reverse inhibitory effect on the WNT/β-Catenin signaling pathway (i.e., activation of PPARγ inhibits the activity of this pathway) and LOX-1 expression (via activation of AMPK and miR-590-5p, or inhibition of ROS and NF-kB). Upstream ERK1/2 hinders the activity of PPARγ. The main PTMs of PPARγ associated with atherosclerosis include acetylation and sumoylation. Acetylation promotes the secretion of IL-6, NOX2, iNOS and other substances responsible for inducing oxidative stress and inflammation. The TZD rosiglitazone has been shown to inhibit PPARγ acetylation. PPARγ sumoylation can be induced by HG/HF/PA via the ROS–IKK–PIAS1 pathway. SUMOylated PPARγ readily interacts with FOXO1, contributing to endothelial IR. Abbreviations: ERK, extracellular regulated protein kinase; PI3K, phosphatidylinositol 3-kinase; NF-kB, nuclear factor kappa B; NOX, NADPH oxidase; rosi, rosiglitazone; RXR, retinoid X receptor; iNOS, inducible NO synthase; IKK, IkB kinase; HG, high glucose; HF, high fat; PA, palmitic acid; PIAS1, protein inhibitor of activated STAT1; PPARγ, peroxisome proliferator-activated receptor γ; oxLDL, oxidized-low-density lipoprotein; LOX-1, lectin-like receptor for oxidized LDL; AMPK, AMP-activated protein kinase; IR, insulin resistance. The red lines represent promotion and the black ones represent inhibition. Meanwhile, the solid lines represent direct action, and the dotted lines represent indirect action, indicating omitted processes.

## Data Availability

Not applicable.

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
