# Peer review of "PPARγ in Atherosclerotic Endothelial Dysfunction: Regulatory Compounds and PTMs"

_ijms, 2023, doi:10.3390/ijms241914494_

Round 1

Reviewer 1 Report (Previous Reviewer 1)

Accepted

Author Response

Thanks for your great comments.

Reviewer 2 Report (New Reviewer)

In this review, the authors aimed to describe the role of PPARγ in the development of AS by listing factors that activate or inhibit it. This would make this factor an important target for disease control. The manuscript must be proofread by a native English speaker. There are many sentences that are unclear and often placed in an inappropriate context. The introduction should be rewritten, especially the first part where the description of AS and its development are not well stated.

1.     Line 23: Endothelial function is essential to atherosclerosis (AS) progression

This sentence should be improved. Put like that, it seems that endothelial function is the cause of the progression of atherosclerosis

2.     The authors in the introduction want to briefly describe the progression of AS. In doing so, however, they put down fragmentary concepts without following a logical thread.

Improve this part of the introduction without overburdening the reading with topics that are not part of the central theme of the review

3.     In the introduction, the authors should talk more about the structure and function of Peroxisome proliferator-activated receptor gamma (PPARγ)

4.     Lines 58-59: To date, the prevention and treatment of atherosclerotic endothelial  disorders have not specifically employed endothelial-targeted approaches [9]

Explain better…..

It does not seem to me that the authors' claim is confirmed by the article [9] where instead the authors claim a beneficial role of statins on endothelial function.....

5.     Line 68: SUMOylation: first write in full ….Small ubiquitin-related modifier (SUMO)

6.     Line 71: Overall, understanding the role of endothelial PPARγ on AS development is of great  importance

This sentence is not in the right place in the text.

7.     Lines 81-84: The endothelium is comprised of ECs lining the innermost  layer of the basal/luminal surface, facilitating fluid flow and substance exchange [26]. Not only blood vessels, but also the kidney, heart, lungs, and other tissues/organs have endothelia with specific structures and functions [27-30].

Very confused and contorted form of writing

8.     Lines 91-93: Analysis of inflammatory factors, oxidation markers, reactive 91 oxygen species (ROS), cell proliferation, and apoptosis also aid in evaluating EC function.

Before discussing the assessment of endothelial function, one should discuss and report on endothelial 'activation', which generally occurs before endothelial dysfunction.

9.     Line 95-97: Albumin concentration, glycocalyx damage, and inflammation all alter the configuration of ECs via intracellular fac-96 tors that change endothelial permeability [38, 39].

Improving the English form. Explain what glycocalyx damage consists of

10. Lines 97-99: Knockdown of endothelial-specific interferon-γ (IFN-γ) receptors prevents vascular barrier disruption caused by injury to the adherens junction protein vascular endothelial (VE)-cadherin [40, 41].

Sentence that does not link conceptually with the text

11. Fig. 1 Barrier integry …..innate immunity?  What does it mean?

I would remove .....others...

Lines 162-163: Endothelial function affects atherosclerosis in many ways,….

It is not endothelial function that acts on AS, but it is the onset of dysfunction that develops the AS. Write down the concepts better!

12. Line 245: Activation of PPARγ may also prompt the opening of maxi-K channels [95]. …… and so what?

13. Line 252: Sigmund et al. employed a mouse model with dominant negative mutations controlled by an endothelial-specific promoter [98, 99]. This endothelium-specific PPARγ interference markedly impaired vasodilatory responses to acetylcholine, which was subsequently restored by a scavenger of superoxide [99]. As such,  overexpression of endothelium-specific PPARγ may prevent endothelial injury in the  aorta by reducing oxidative stress, rather than by blunting NF-κB activity [98].

Unclear text. What is the role of PPARγ in this case?

14. Line 367:….. increasing the production of NO, and improving PPARγ-dependent dilation.  Dilation of what?

15. In this review, the authors did not report an important topic, namely what role PPARγ plays in lectin-like oxidised low density lipoprotein LDL receptor-1 (LOX-1) which is considered to be an important factor in vascular damage and is mainly involved in cardiovascular risk. I think it is important to discuss this topic.

The manuscript must be proofread by a native English speaker.

Author Response

Thanks for your great comments. Our point to point responses are in the attached file. I really appreciate it.

Round 2

Reviewer 2 Report (New Reviewer)

Comments:

The function and structure of PPARy described is too long. Try to summarise it as much as possible.

Lines 64-65: Improve the sentence

There are still language inaccuracies, for example “The endothelium synthesizes…..”

would be better without the article

Line 104: Few studies have thoroughly investigated the role of PPARγ in atherosclerotic endothelial dysfunction. For example, a review by Gimbrone Jr. and García-Cardeña evaluated the mechanisms of atherosclerosis and the significant relationship of endothelial dysfunction with other atherosclerosis risk factors, but described only a limited role for PPARγ.

The authors report a study without indicating what role is described for PPARγ

Lines 239-240: Endothelial dysfunction plays a key role in the formation of atherosclerosis, but the underlying cellular and molecular mechanisms remain unclear.

A statement I cannot accept!

Minor editing of English language required

Author Response

The authors greatly appreciate your great suggestions and thorough consideration of our manuscript entitled “PPARγ in atherosclerotic endothelial dysfunction: regulatory compounds and PTMs”. Our point-by-point responses to each comment are presented in our attached file. All changes in the revised manuscript are highlighted in red.

Round 3

Reviewer 2 Report (New Reviewer)

I must admit that the authors made many inaccuracies and initially the manuscript was so inadequate and confusing that it almost had to be rejected. However, with a great deal of effort the authors managed to improve the manuscript and it is now sufficiently improved to justify publication in IJMS.

Minor editing of English language required.

This manuscript is a resubmission of an earlier submission. The following is a list of the peer review reports and author responses from that submission.

Round 1

Reviewer 1 Report

Major corrections required.

Few lines need attention to make it interesting.

Author Response

Response: The authors greatly appreciate your thoughtful insights and thorough consideration of our manuscript entitled “PPARγ on atherosclerotic endothelial dysfunction: focus on compounds and PTMs”. Our point-by-point responses to each comment are presented below. All changes in the revised manuscript are highlighted in red.

Reviewer 2 Report

The subject of this review is the role of PPARgamma on atherosclerotic endothelial dysfunction with a focus on compounds and PTMs.PPARg is a nuclear receptor that plays a critical role in glucose homeostasis and adipocity but its roles in the endothelium are not well studied. With this review the authors are attempting to fill this gap.

Major issues:

1. The authors spend a significant fraction of the paper (almost 5 pages) on general information about the function and the dysfunction of the endothelium.This section needs significant trimming

2. A section of the structure of PPARg and the role of each domain is missing on the manuscript. This is essentil for a better understanding the PTMs of PPARg

3. Some compounds have no reported roles on the endothelium (i.e. phenolic compounds, alkaloids) and should be removed from the text

4. The authors should focus only on the PPARg PTMs that have an effect on endothelial cells.

5. Τhe manuscript needs extensive editing and optimization language

Τhe manuscript needs extensive editing and optimization language

Author Response

Response: The authors greatly appreciate your thoughtful insights and thorough consideration of our manuscript entitled “PPARγ on atherosclerotic endothelial dysfunction: focus on compounds and PTMs”. Our point-by-point responses to each comment are presented below. All changes in the revised manuscript are highlighted in red.

Comments and Suggestions for Authors

The subject of this review is the role of PPARgamma on atherosclerotic endothelial dysfunction with a focus on compounds and PTMs.PPARg is a nuclear receptor that plays a critical role in glucose homeostasis and adipocity but its roles in the endothelium are not well studied. With this review the authors are attempting to fill this gap.

Major issues:

  1. The authors spend a significant fraction of the paper (almost 5 pages) on general information about the function and the dysfunction of the endothelium. This section needs significant trimming.

Response: Thanks for your great suggestion. We have reorganized the first section and have made it more concise.

  1. A section of the structure of PPARg and the role of each domain is missing on the manuscript. This is essentil for a better understanding the PTMs of PPARγ

Response:  This is a good point. We have described the structure of PPARγ and the function of each domain in section 3.1.1.

  1. Some compounds have no reported roles on the endothelium (i.e. phenolic compounds, alkaloids) and should be removed from the text

Response:  Thanks for this great suggestion. We checked all compounds carefully again and have corrected all of them. Phenolic compounds contain Resveratrol and Hydroxytyrosol. Resveratrol promotes cholesterol efflux from macrophages in a PPARγ-dependent manner while in endothelial cells it was not yet reported. Hydroxytyrosol doesn’t have reported roles on the endothelium. So, we excluded the compounds mentioned above. We also excluded the PDTC and N15. On the contrary, alkaloids (Hypaphorine and berberine) can work on atherosclerosis in a PPARγ-dependent way in endothelium. So, we decided to keep it. The corresponding compounds in the table 1 had been removed with no revision traces due to the deletion of the entire line.

  1. The authors should focus only on the PPARg PTMs that have an effect on endothelial cells.

Response: We removed the phosphorylation of PPARg since few studies mention its function in endothelial cells. Since its selective activity of PPARg by phosphorylation, its potential roles in endothelial cells worthy further investigation, which we mention in the discussion part of new version.

  1. Τhe manuscript needs extensive editing and optimization language

Response: Thanks for your great suggestion. We have got through and revise it thoroughly and try to make it more concise and clear.

Round 2

Reviewer 1 Report

Accepted